# Stabilization of CDK6 by ribosomal protein uS7, a target protein of the natural product fucoxanthinol

Yosuke Iizumi [1✉], Yoshihiro Sowa[1], Wakana Goi[1], Yuichi Aono[1,5], Motoki Watanabe[1], Yoichi Kurumida[2], Tomoshi Kameda [2], Kenichi Akaji[3], Masatoshi Kitagawa[4] & Toshiyuki Sakai[1,6]

Cyclins and cyclin-dependent kinases (CDKs) regulate the cell cycle, which is important for cell proliferation and development. Cyclins bind to and activate CDKs, which then drive the cell cycle. The expression of cyclins periodically changes throughout the cell cycle, while that of CDKs remains constant. To elucidate the mechanisms underlying the constant expression of CDKs, we search for compounds that alter their expression and discover that the natural product fucoxanthinol downregulates CDK2, 4, and 6 expression. We then develop a method to immobilize a compound with a hydroxyl group onto FG beads® and identify human ribosomal protein uS7 (also known as ribosomal protein S5) as the major fucoxanthinol-binding protein using the beads and mass spectrometry. The knockdown of uS7 induces G1 cell cycle arrest with the downregulation of CDK6 in colon cancer cells. CDK6, but not CDK2 or CDK4, is degraded by the depletion of uS7, and we furthermore find that uS7 directly binds to CDK6. Fucoxanthinol decreases uS7 at the protein level in colon cancer cells. By identifying the binding proteins of a natural product, the present study reveals that ribosomal protein uS7 may contribute to the constant expression of CDK6 via a direct interaction.

[1] Department of Molecular-Targeting Prevention, Graduate School of Medical Science, Kyoto Prefectural University of Medicine, Kawaramachi-Hirokoji, Kamigyo-ku, Kyoto 602-8566, Japan. [2] Artificial Intelligence Research Center, National Institute of Advanced Industrial Science and Technology (AIST), Tokyo 135-0064, Japan. [3] Department of Medicinal Chemistry, Kyoto Pharmaceutical University, Yamashina-ku, Kyoto 607-8412, Japan. [4] Department of Molecular Biology, Hamamatsu University School of Medicine, 1-20-1 Handayama, Higashi-ku, Hamamatsu, Shizuoka 431-3192, Japan. [5] Present address: Department of Biomedical Chemistry, Graduate School of Science and Technology, Kwansei Gakuin University, 2-1 Gakuen, Sanda, Hyogo 669-1337, Japan. [6] Present address: Department of Drug Discovery Medicine, Graduate School of Medical Science, Kyoto Prefectural University of Medicine, Kawaramachi-Hirokoji, Kamigyo-ku, Kyoto 602-8566, Japan. ✉email: yiizumi@koto.kpu-m.ac.jp

The cell cycle of organisms is essential for cell proliferation and development and is controlled by cyclins and cyclin-dependent kinases (CDKs)[1]. In the G1 phase of the cell cycle, the CDK2-cyclin E and CDK4/6-cyclin D complexes phosphorylate the tumor suppressor RB protein, resulting in its inactivation and the acceleration of cancer cell proliferation[2]. The expression of cyclins periodically changes throughout the cell cycle, because it is regulated by the ubiquitin-proteasome system[3]. On the other hand, the expression of CDKs remains constant[4], which enables rapid responses to changes in the expression of cyclins. The molecular chaperone Hsp90, which stabilizes >50% of human kinases, has been shown to bind to and stabilize CDK4[5,6]. However, the mechanisms by which the expression of CDKs is maintained have not yet been elucidated in detail.

Ribosomal proteins are components of the ribosome, which governs protein translation; however, a number of ribosomal proteins are known to function outside of the ribosome[7]. Ribosomal protein uL5 directly binds to and inhibits the E3 ligase HDM2, resulting in the stabilization of p53[8]. Furthermore, ribosomal protein uS7 has been implicated in hepatic fibrosis[9] and osteoclastogenesis[10]. In the regulation of the cell cycle, ribosomal proteins uS15 and uL3 have been shown to positively and negatively regulate the G1/S transition by downregulating p27[Kip1] and cyclin D1, respectively[11,12]. Moreover, a recent study reported that ribosomal protein uS11 accumulated in the course of cellular senescence, directly inhibited the kinase activity of CDK4, and induced senescence[13]. Since ribosomal proteins are comparatively abundant, they may play important roles other than protein translation.

FG beads[®] are magnetic nano-carriers[14] produced by improving high-performance affinity beads for the purification of drug receptors[15]. These beads have been used to elucidate the mechanisms of action of various chemical compounds. Cereblon (CRBN) was previously identified as a primary target of thalidomide, which induces teratogenicity, and a substrate receptor of the E3 ubiquitin ligase complex CRL4[CRBN] using FG beads[®][16]. Furthermore, lenalidomide, a thalidomide derivative, was effective against multiple myeloma and myelodysplastic syndrome with the deletion of chromosome 5q by degrading IKZF1, IKZF3[17], and CK1α[18] through CRL4[CRBN]. FG beads[®] are valuable for elucidating novel biological mechanisms as well as the mechanisms of action of chemical compounds.

To clarify the mechanisms contributing to the constant expression of CDKs, we initially searched for compounds that affect the expression of CDKs, and found that the natural compound fucoxanthinol downregulated their expression. We then identified human ribosomal protein uS7 as a fucoxanthinol-binding protein using FG beads[®], and showed that it was involved in the protein stability of CDK6. Using fucoxanthinol, the present study revealed the stabilization of CDK6 by ribosomal protein uS7.

## Results

### The natural product fucoxanthinol downregulates CDK2, 4, and 6 expression.

To elucidate the mechanisms underlying the constant expression of CDK2, 4, and 6, we initially searched for compounds that induce G1 cell cycle arrest in human colon cancer HT-29 and SW480 cells. As shown in Fig. 1a and Supplementary Fig. 1a, the natural product fucoxanthinol inhibited the proliferation of HT-29 and SW480 cells, and the effective concentrations of fucoxanthinol against SW480 cells were lower than those against HT-29 cells. At the effective concentrations, fucoxanthinol induced G1 cell cycle arrest in HT-29 and SW480 cells in a concentration-dependent manner (Fig. 1b and Supplementary Fig. 1b). A high concentration of fucoxanthinol (20 or

10 μM in HT-29 or SW480 cells, respectively) also induced G2/M arrest. Fucoxanthinol dephosphorylated the RB protein at Ser780 and Ser807/811, the corresponding kinases CDK2, 4, and 6[19] were downregulated, and the CDK inhibitor p21 was induced in HT-29 cells (Fig. 1c). Fucoxanthinol downregulated CDK2, 4, and 6 expression also in SW480 cells (Supplementary Fig. 1c). It has been reported that fucoxanthinol decreases the expression levels of CDK4 and 6 in human T-cell leukemia virus type 1-infected T-cell lines[20]. These findings indicate that fucoxanthinol is useful for analyzing the regulation of CDK expression.

### Ribosomal protein uS7 is the major direct binding protein of fucoxanthinol.

We attempted to identify proteins that directly bind to fucoxanthinol, because they may contribute to the constant expression of CDKs. To purify fucoxanthinol-binding proteins with magnetic affinity beads (FG beads[®]), we developed a method in which a compound with a hydroxyl group was conjugated to the carboxyl group on FG beads[®]. Fucoxanthinol was covalently immobilized onto the beads with 4-dimethylaminopyridine (DMAP), as shown in Fig. 2a. We then purified fucoxanthinol-binding proteins from HT-29 whole-cell lysates and detected these proteins by silver staining (Fig. 2b). One major binding protein was found and identified as human ribosomal protein uS7 by mass spectrometry (Fig. 2c). As shown in Fig. 2d, the binding of fucoxanthinol to uS7 was also confirmed by western blotting. The recombinant His-tagged uS7 protein (His-uS7) bound to fucoxanthinol-fixed beads (Fig. 2e), indicating that fucoxanthinol directly bound to the uS7 protein.

Furthermore, the microscopic state of the interaction between uS7 and fucoxanthinol was examined using docking and molecular dynamics (MD) simulations. We generated 100 poses of the complex of uS7 and fucoxanthinol by docking simulations (Fig. 3a). In the top 2 poses, fucoxanthinol bound to a similar position on uS7, and the allenic group of fucoxanthinol was located at a hydrophobic pocket of uS7 (Fig. 3b). To investigate the stability of the complex structure, we further performed MD simulations. One hundred ns simulations were performed three times. In all simulations, fucoxanthinol remained at the same position of uS7 (Supplementary Movie 1), suggesting that it strongly bound to the hydrophobic pocket of uS7.

### Knockdown of uS7 downregulates CDK6 protein expression in human colon cancer cell lines.

Although uS7 is a component of the ribosome, limited information is currently available on its function in the proliferation of cancer cells. To investigate the role of uS7 in cancer cells, we performed the knockdown of uS7 in HT-29 cells (Fig. 4a). The knockdown of uS7 inhibited the proliferation of HT-29 cells (Fig. 4b) and induced G1 cell cycle arrest (Fig. 4c). The dephosphorylation of the RB protein and downregulation of CDK2, 4, and 6 protein expression were observed in uS7-depleted HT-29 cells (Fig. 4d), which was consistent with the effects of the fucoxanthinol treatment (Fig. 1c). In contrast to the treatment with fucoxanthinol, the depletion of uS7 did not induce p21. A subcellular fractionation analysis showed that CDK2, 4, and 6 protein expression in both the nuclear and cytosolic fractions was downregulated by the depletion of uS7 (Supplementary Fig. 2). On the other hand, the knockdown of uS7 only downregulated CDK6 protein expression in SW480 cells (Fig. 4e). These results suggest that uS7 regulates CDK6 protein expression in human colon cancer cell lines, suggesting a specific relationship between uS7 and CDK6.

We also examined the effects of the knockdown of uS7 on other ribosomal proteins. As shown in Supplementary Fig. 3a, the knockdown of uS7 slightly downregulated the expression of uS4 (a component of the small ribosomal subunit) and uL3 (a component of the large ribosomal subunit) in HT-29 cells. On the other hand, the

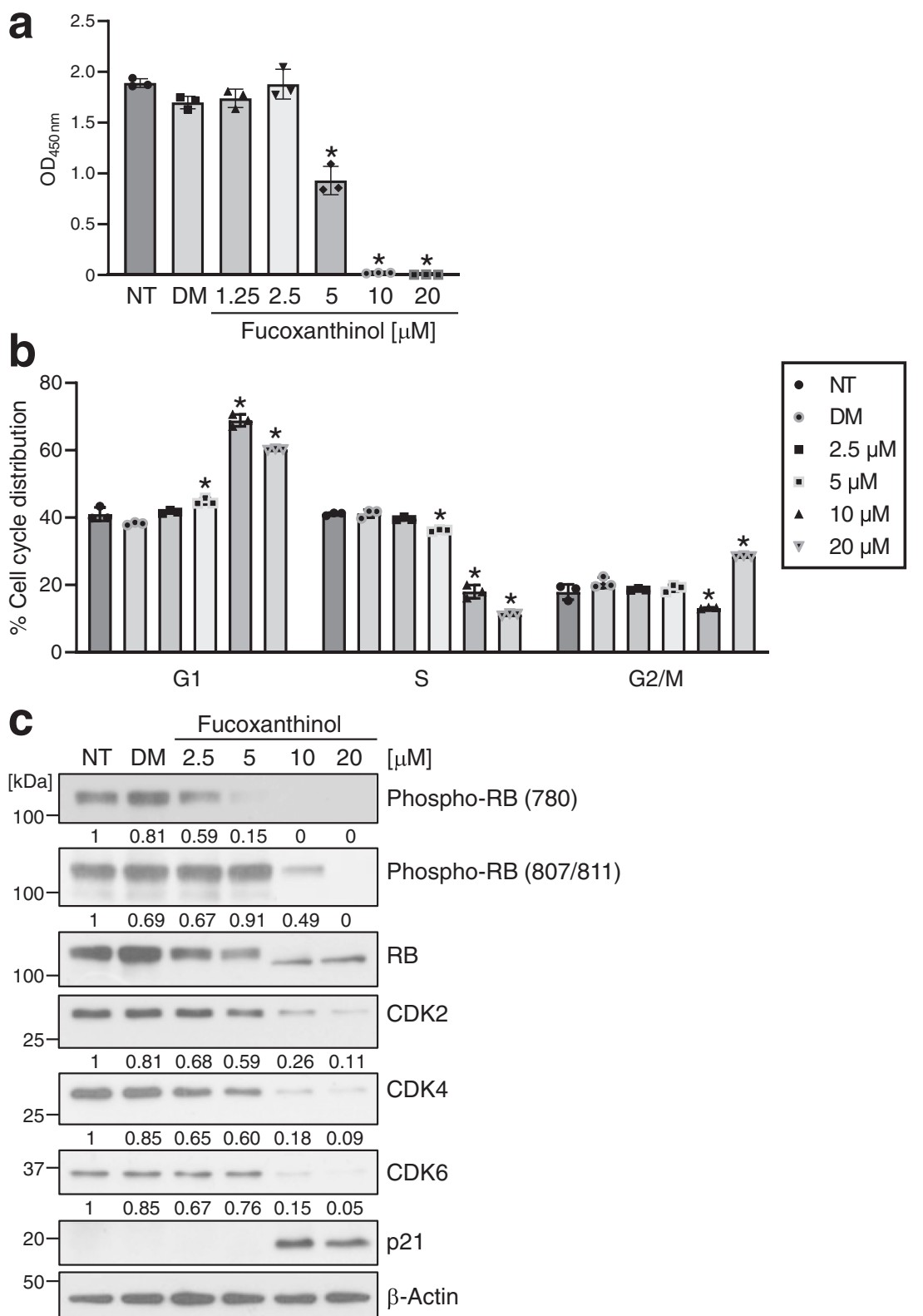

**Fig. 1 Fucoxanthinol induces G1 cell cycle arrest with the downregulation of CDK2, 4, and 6 protein expression. a** Human colon cancer HT-29 cells were treated with the indicated doses of fucoxanthinol for 72 h. The proliferation of cells was measured using the CCK-8 assay. **b** HT-29 cells were treated with fucoxanthinol for 24 h, and the cell cycle was analyzed by flow cytometry. **c** A western blot analysis of HT-29 cells treated with fucoxanthinol for 24 h. The signal of each western blot was quantified using ImageJ software (Version 1.52a) and normalized by the value of β-actin. The value of each signal was indicated below the blot. NT: non-treated, DM: 0.1% DMSO, Data are means ± S.D. ($n = 3$ biologically independent experiments). *$P < 0.05$ significantly different from NT.

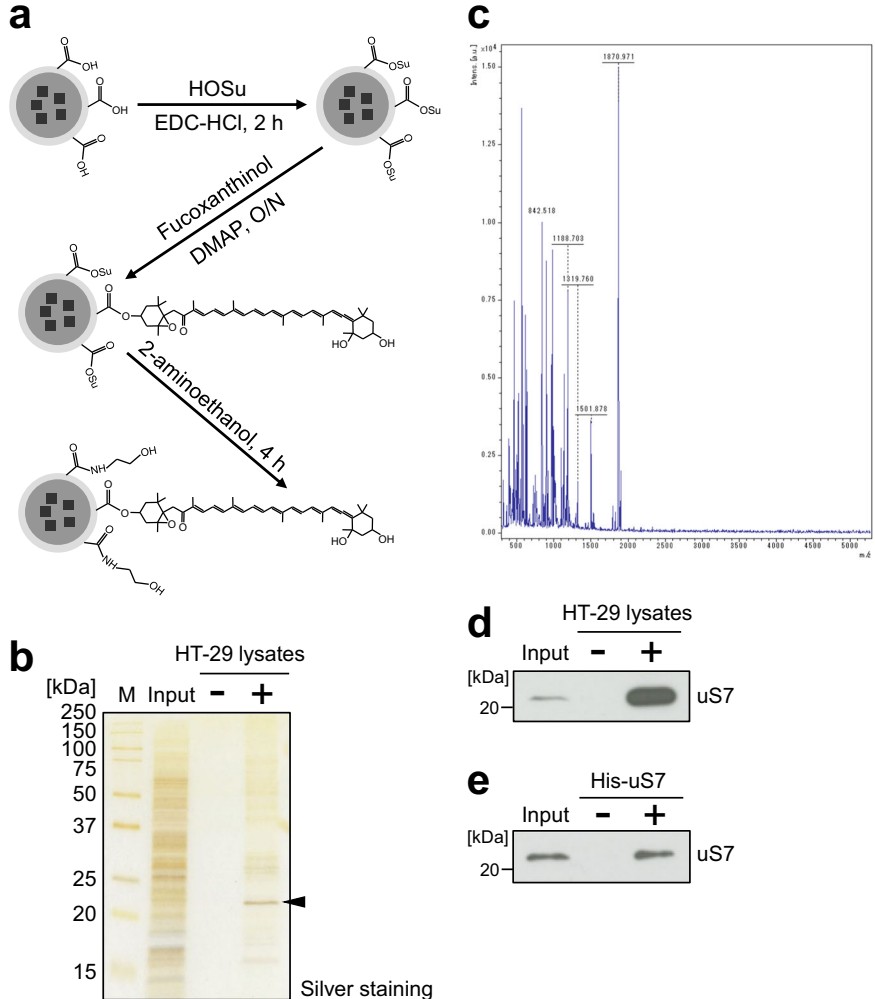

**Fig. 2 Ribosomal protein uS7 is the major binding protein of fucoxanthinol. a** The scheme for the immobilization of fucoxanthinol onto FG beads® with carboxyl groups. **b** Whole-cell extracts of HT-29 cells were incubated with empty (−) or fucoxanthinol-fixed (+) FG beads®, and the binding proteins of fucoxanthinol were purified and detected by silver staining. **c** The major binding protein indicated by an arrowhead in **b** was analyzed by mass spectrometry, and the molecular masses of the peptides detected are shown. **d** Whole-cell extracts of HT-29 cells were incubated with fucoxanthinol-fixed beads, and the binding of uS7 was detected by western blotting. **e** The recombinant His-tagged uS7 protein (His-uS7) was incubated with fucoxanthinol-fixed beads, and the binding of uS7 was detected by western blotting. HOSu: *N*-hydroxysuccinimide, EDC-HCl: 1-ethyl-3-(3-dimethylaminopropyl)-carbodiimide hydrochloride, DMAP: 4-dimethylaminopyridine.

knockdown of uS7 downregulated the expression of uS4, but not uL3 in SW480 cells (Supplementary Fig. 3b). Therefore, the knockdown of uS7 may influence the expression of other components of the small ribosomal subunit in a cell line-specific manner.

**uS7 contributes to the stability of the CDK6 protein**. To examine the role of uS7 in CDK2, 4, and 6 protein expression, we assessed the stability of these proteins using a cycloheximide chase assay. After an incubation with cycloheximide for 4 h, the degradation of the CDK6 protein was more rapid in uS7-depleted HT-29 cells than in non-depleted HT-29 cells (Fig. 5a, b), whereas degradation rates of the CDK2 and 4 proteins were not largely altered in uS7-depleted HT-29 cells. In SW480 cells, the degradation of the CDK6 protein was also more rapid in uS7-depleted cells than in non-depleted cells (Supplementary Fig. 4a, b). The proteasome inhibitor MG-132 suppressed reductions in the CDK6 protein by the depletion of uS7 (Fig. 5c), suggesting that the depletion of uS7 decreased CDK6 expression levels via the ubiquitin-proteasome pathway. These results suggest that uS7 maintains the constant expression of the CDK6 protein.

**uS7 directly binds to CDK6**. We investigated the mechanisms by which uS7 regulated the stability of the CDK6 protein. A co-immunoprecipitation analysis showed that uS7 interacted with CDK6, but not CDK2, CDK4, cyclin D, or cyclin E (Fig. 5d), suggesting that uS7 binds to CDK6. A glutathione *S*-transferase (GST) pull-down analysis also showed that the recombinant Myc-DDK-tagged CDK6 protein bound to GST-fused uS7 (GST-uS7), but not to GST alone (Fig. 5e). These results indicate that uS7 directly binds to CDK6, and this interaction may contribute to the stabilization of CDK6 by uS7.

**Fucoxanthinol reduces the protein level of uS7, leading to G1 cell cycle arrest**. We also examined the effects of fucoxanthinol on uS7. As shown in Fig. 6a, fucoxanthinol downregulated uS7 protein expression for 6 h. However, it did not downregulate uS7 mRNA expression for 6 h (Fig. 6b), indicating that fucoxanthinol reduced uS7 at the protein level. Although fucoxanthinol induced G1 cell cycle arrest in non-depleted cells, the induction of G1 cell cycle arrest by fucoxanthinol was weaker in uS7-depleted cells

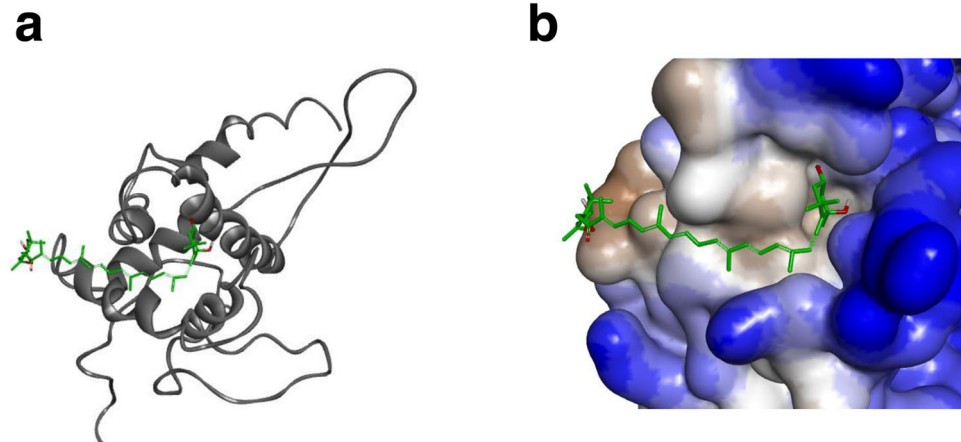

**Fig. 3 The molecular binding model for the interaction between uS7 and fucoxanthinol. a** The top-ranked pose in the docking simulation. Fucoxanthinol (green stick) and uS7 (gray cartoon) are shown. **b** Hydrophobicity around the pocket to which fucoxanthinol binds. Brown and blue colors correspond to the hydrophobic and hydrophilic regions of uS7, respectively. Images (**a**, **b**) were made using BIOVIA Discovery Studio 2018 software (Dassault Systèmes, Vélizy-Villacoublay, France).

(Fig. 6c). These results suggest that fucoxanthinol induces G1 cell cycle arrest by binding to and decreasing the uS7 protein.

## Discussion
To elucidate the mechanisms maintaining the constant expression of CDKs, we initially searched for compounds that affect the expression of CDKs and found that the natural product fucoxanthinol reduced CDK2, 4, and 6 at the protein level (Fig. 1c). We then identified uS7 as the primary binding protein of fucoxanthinol (Fig. 2b, c), and showed that it was involved in the stability of the CDK6 protein (Fig. 5 and Supplementary Fig. 4). These results raise the possibility that ribosomal protein uS7 contributes to the constant expression of CDK6. On the other hand, since the treatment with cycloheximide rapidly degraded the CDK4 protein (Fig. 5a, b), this protein may be continually degraded and resynthesized. Further studies on uS7 will provide a more detailed understanding of the mechanisms underlying the stability of CDK6.

The depletion of uS7 downregulated CDK2, 4, and 6 expression in HT-29 cells (Fig. 4d), whereas only CDK6 was reduced in uS7-depleted SW480 cells (Fig. 4e). Since uS7 plays a role in the initiation of translation as a component of the ribosome[21], we considered CDK2 and 4 expression to be downregulated in uS7-depleted HT-29 cells due to the partial inhibition of translation and also suggested that uS7 does not play a role in translation in SW480 cells. Some of the functions of ribosomal proteins depend on the p53 status[22]; therefore, differences in the regulation of CDK2 and 4 may also be attributed to the different status of p53 between HT-29 and SW480 cells, which carry the dominant-negative mutation (R273H)[23] and mutations (R273H/P309S) retaining some of the normal functions of p53[24], respectively. On the other hand, the depletion of uS7 promoted the degradation of the CDK6 protein in both human colon cancer cell lines (Fig. 5 and Supplementary Fig. 4), and we observed a direct interaction between uS7 and CDK6 (Fig. 5e). These results suggest an extraribosomal function for uS7, which stabilizes the CDK6 protein by this direct interaction.

The uS7 protein was involved in the protein stability of CDK6 (Fig. 5 and Supplementary Fig. 4). A previous study reported that the CDK6 protein was modified by small ubiquitin-like modifier 1 (SUMO1) and stabilized through the prevention of the ubiquitin-mediated degradation of CDK6[25]. Since the downregulated expression of the CDK6 protein via the depletion of uS7 was dependent on the ubiquitin-proteasome system (Fig. 5c), the depletion of uS7 may destabilize CDK6 by removing SUMO1.

Therefore, further studies on the relationship between uS7 and SUMO1 modifications are needed.

The knockdown of uS7 downregulated CDK6 protein expression in both the nuclear and cytoplasmic fractions (Supplementary Fig. 2). The CDK6-cyclin D complex phosphorylates the RB protein in the nucleus. However, the roles of CDK6 in the cytoplasm currently remain unclear. A previous study demonstrated that CDK6 formed a complex with STAT3 or c-Jun to regulate *p16INK4a* or *VEGF-A* expression at the transcriptional level[26]. Since CDK6 also possesses this kinase-independent function, the cytoplasm may have an important role as a storage for an additional source of the CDK6 protein.

A high concentration of fucoxanthinol induced G2/M arrest as well as G1 arrest (Fig. 1b and Supplementary Fig. 1b). A previous study reported that a high concentration of fucoxanthin, the precursor of fucoxanthinol, downregulated the expression of cyclin B1, a positive regulator of the G2/M transition;[27] therefore, a high concentration of fucoxanthinol may also induce G2/M arrest by downregulating the expression of cyclin B1. When G2 arrest is induced, because the cells which are in at least the S and G2 phases must be arrested in the G2/M phase, a high concentration of fucoxanthinol may induce both G1 and G2/M arrest.

Collectively, the present results suggest that fucoxanthinol induces G1 cell cycle arrest by directly binding to and reducing ribosomal protein uS7. We previously reported that the natural product apigenin induced G2/M cell cycle arrest by directly binding to ribosomal protein uS4[28]. On the other hand, ribosomal protein uL3 was shown to prevent S cell cycle arrest induced by the natural compound, S-adenosyl-L-methionine[29]. These findings imply relationships between cell cycle regulation by ribosomal proteins and the bioactivities of natural products.

The present study focused on the constant expression of CDKs, while alterations in CDK4/6 expression are attracting increasing attention because the CDK4/6 inhibitors palbociclib, ribociclib, and abemaciclib have exhibited clinical efficacy against advanced breast cancer[30–33] and have been approved for clinical use. In a genomic analysis of breast cancers that were resistant to CDK4/6 inhibitors, loss-of-function mutations were found in the *FAT1* gene, and the knockout of *FAT1* conferred resistance to CDK4/6 inhibitors by the overexpression of the CDK6 protein via the suppression of the Hippo pathway[34]. Therefore, it is also valuable for studies on cancer therapies to investigate the regulation of CDK4/6 expression.

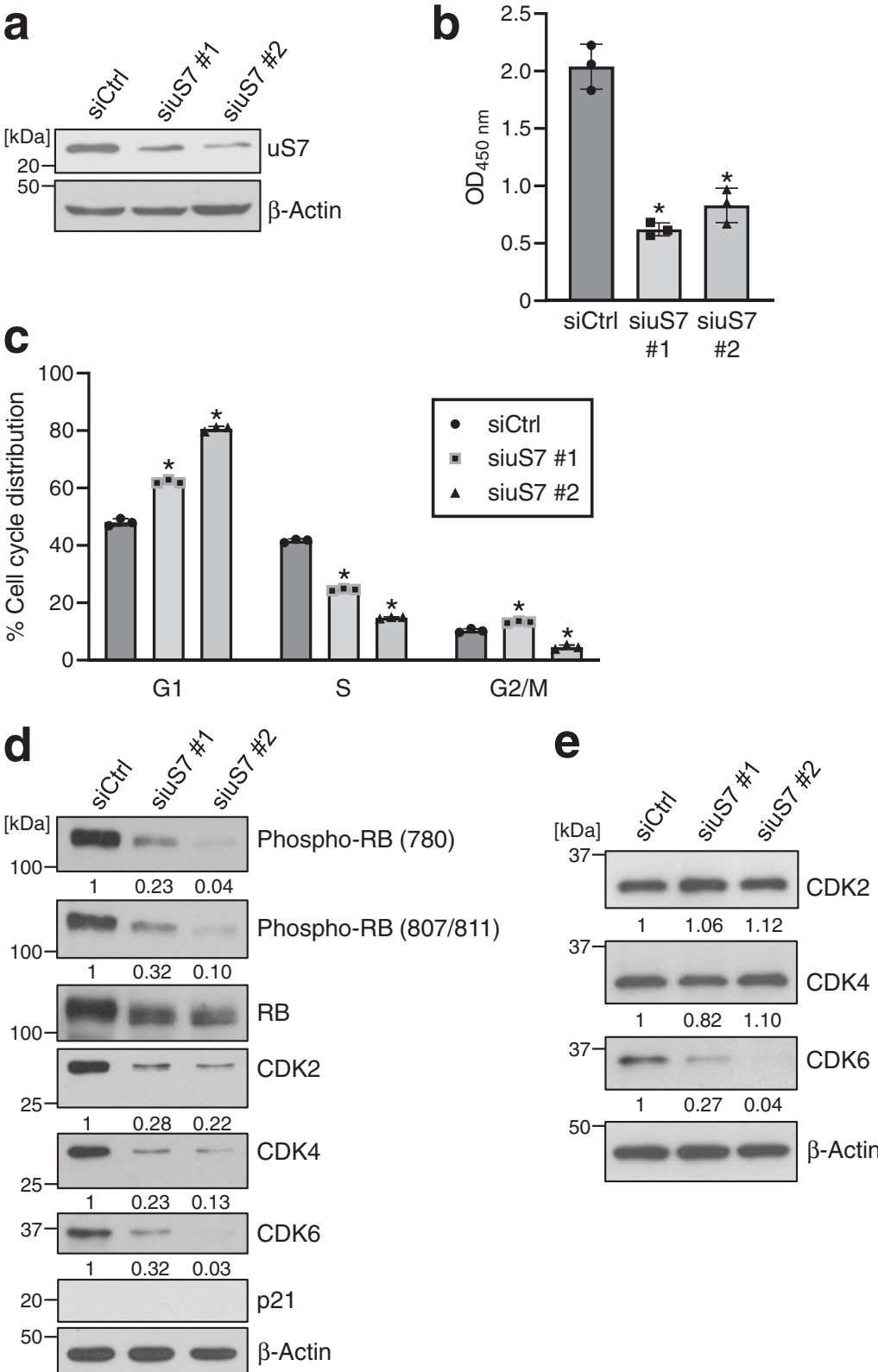

**Fig. 4 The knockdown of uS7 reduces the CDK6 protein in colon cancer cell lines. a–d** HT-29 cells were transfected with negative control siRNA (siCtrl) or two different siRNAs targeting uS7 (siuS7 #1 and #2). After 48 h, the depletion of uS7 was confirmed (**a**). After 72 h, cell proliferation was examined using the CCK-8 assay (**b**). After 48 h, the cell cycle was analyzed by flow cytometry (**c**), and protein levels were investigated by western blotting (**d**). **e** SW480 cells were transfected with each siRNA. After 48 h, cells were lysed and subjected to western blotting. In **d**, **e**, the signal of each western blot was quantified using ImageJ software and normalized by the value of β-actin. The value of each signal was indicated below the blot. Data are means ± S.D. ($n = 3$ biologically independent experiments). $*P < 0.05$ significantly different from siCtrl.

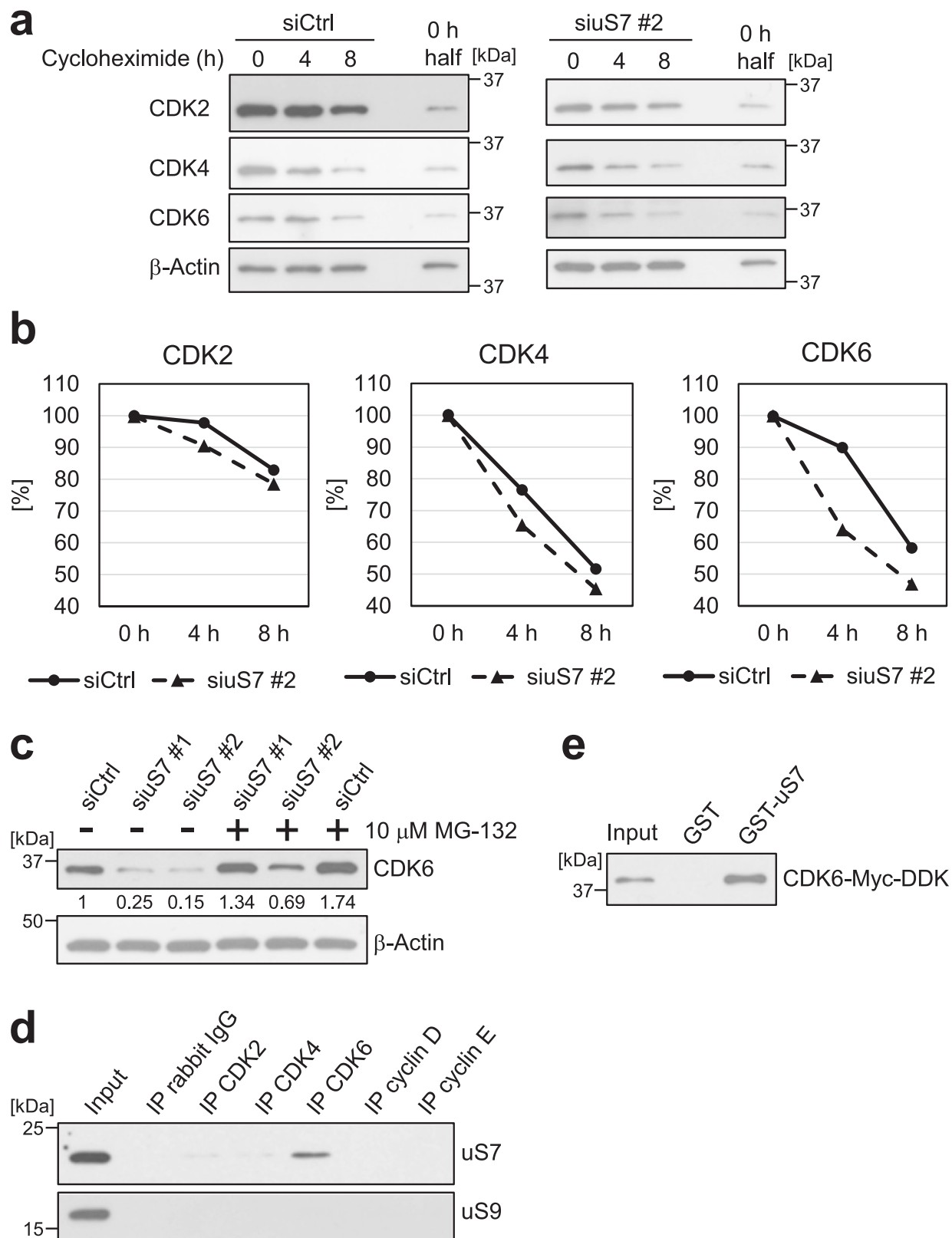

## Methods

**Cell culture**. Human colon cancer HT-29 cells were obtained from the NCI-60 cancer cell line panel of the National Cancer Institute Developmental Therapeutics Program. Human colon cancer SW480 cells were obtained from the American Type Culture Collection. These cells were maintained in Dulbecco's modified Eagle's medium (DMEM) supplemented with 10% fetal bovine serum

(FBS), 4 mM glutamine, 50 U/ml penicillin, and 100 μg/ml streptomycin at 37 °C in 5% $CO_2$.

**Reagents**. Fucoxanthinol was purchased from FUJIFILM Wako Pure Chemical Corporation (Osaka, Japan). Cycloheximide and Phosphatase Inhibitor Cocktail

**Fig. 5 uS7 directly binds to CDK6 and is involved in the stability of the CDK6 protein. a** HT-29 cells were transfected with siCtrl or siuS7 #2. After 18 h, cells were treated with 20 μg/ml cycloheximide and lysed at the indicated times. The expression levels of CDK2, 4, and 6 were analyzed by western blotting. The sample, designated as "0 h half", is identical to half the amount of the sample at 0 h. **b** The expression levels of CDK2, 4, and 6 at each time point were quantified. The expression level at 0 h was defined as 100%. **c** HT-29 cells were transfected with siCtrl, siuS7 #1, or siuS7 #2. After 24 h, cells were incubated with 10 μM MG-132 for 24 h, lysed, and subjected to western blotting. The signal of each western blot was quantified using ImageJ software and normalized by the value of β-actin. The value of each signal was indicated below the blot. **d** HT-29 cells were lysed, and the endogenous immunoprecipitation assay was performed with anti-CDK2, CDK4, CDK6, cyclin D, and cyclin E antibodies, respectively. Ribosomal proteins uS7 and uS9 in immunoprecipitates were detected by western blotting. **e** A GST pull-down assay was performed by incubating the recombinant Myc-DDK-tagged CDK6 protein with GST or GST-fused uS7 (GST-uS7)-bound beads. The CDK6 protein that bound to these beads was detected by western blotting.

(EDTA free) were purchased from Nacalai Tesque (Kyoto, Japan). MG-132 was purchased from Peptide Institute (Osaka, Japan).

**Cell proliferation assay.** The proliferation of treated HT-29 or SW480 cells was examined using Cell Counting Kit-8 (CCK-8, Dojindo Molecular Technologies, Kumamoto, Japan). The CCK-8 solution was added to the medium of the cells. After 4 h, the absorbance (450 nm) of the samples was measured using Multiskan FC (Thermo Fisher Scientific, Waltham, MA, USA).

**Cell cycle analysis.** Treated HT-29 or SW480 cells were harvested by trypsinization. After centrifugation at $780 \times g$ for 5 min at room temperature, the cells were suspended in phosphate-buffered saline containing 0.1% Triton X-100, 150 μg/ml RNase A, and 5 μg/ml propidium iodide. The suspension was filtered through a mesh sheet (Kurabo Industries Ltd., Osaka, Japan). The DNA content in the stained nuclei was measured by BD FACSCalibur (Becton, Dickinson and Company, Franklin Lakes, NJ, USA) and BD CellQuest Pro software (version 6.0; Becton, Dickinson and Company). No gating was used, and 10,000 events were recorded for each sample. The cell cycle distribution of the cells was analyzed using ModFit LT software (version 3.3.11; Verity Software House, Inc., Topsham, ME, USA).

**Western blot analysis.** Treated HT-29 or SW480 cells were lysed with RIPA buffer (50 mM Tris-HCl [pH 8.0], 150 mM NaCl, 1% NP-40, 0.5% deoxycholic acid, 0.1% SDS, 1 mM DTT, 0.5 mM PMSF, and Phosphatase Inhibitor Cocktail) at 4 °C for 30 min and centrifuged at $20,400 \times g$ for 10 min at 4 °C. The proteins in the supernatants were subjected to SDS-PAGE and transferred to Immobilon-P membranes (Millipore, Billerica, MA, USA). The membranes were incubated with Tris-buffered saline containing 5% skim milk at 4 °C overnight and then incubated with each primary antibody at room temperature for 1 h. The primary antibodies used in the present study were anti-phospho-RB-Ser780 (#9307, Cell Signaling Technology, Danvers, MA, USA, 1:200 dilution), phospho-RB-Ser807/811 (#9308, Cell Signaling Technology, 1:200 dilution), RB (Cat. No. 554136, BD Biosciences, San Jose, CA, USA, 1:200 dilution), CDK2 (sc-163, Santa Cruz Biotechnology, Dallas, TX, USA, 1:2000 dilution), CDK4 (sc-601, Santa Cruz Biotechnology, 1:1000 dilution), CDK6 (sc-177, Santa Cruz Biotechnology, 1:500 dilution), p21 (sc-397, Santa Cruz Biotechnology, 1:500 dilution), β-actin (A5441, Sigma-Aldrich, St. Louis, MO, USA, 1:2000 dilution), Lamin A/C (#2032, Cell Signaling Technology, 1:500 dilution), uS7 (ab58345 or ab210745, Abcam, Cambridge, UK, 1:500 or 1:1000 dilution), uS9 (ab26159, Abcam, 1:500 dilution), uS4 (ab74711, Abcam, 1:1000 dilution), uL3 (ab241412, Abcam, 1:500 dilution), and DDK (TA50011-100, OriGene Technologies, Rockville, MD, USA, 1:500 dilution). The membranes were then incubated with HRP-linked anti-rabbit IgG (NA934V, GE Healthcare, Chicago, IL, USA, 1:2000 dilution) or HRP-linked anti-mouse IgG (NA931V, GE Healthcare, 1:2000 dilution) at room temperature for 1 h. Each protein was detected on BioMax XAR film (Carestream Health, Inc., Rochester, NY, USA) using Chemi-Lumi One L (Nacalai Tesque) or Immobilon Western (Millipore).

**Quantitative reverse transcription-PCR (qRT-PCR) analysis.** Total cellular RNA was extracted from treated HT-29 cells using Sepasol-RNA I Super G (Nacalai Tesque), and complementary DNA (cDNA) was synthesized from total RNA using High-Capacity cDNA Reverse Transcription kit (Applied Biosystems, Foster City, CA, USA). A qRT-PCR analysis was performed using the StepOnePlus Real-Time PCR system (Applied Biosystems) with TaqMan probes (Applied Biosystems) for human uS7 (Hs00734849_g1) and human GAPDH (Hs02758991_g1).

**Immobilization of fucoxanthinol onto FG beads®.** FG beads® with carboxyl linkers (TAS8848N1140, Tamagawa Seiki, Nagano, Japan) were incubated with *N*-hydroxysuccinimide (Peptide Institute) and 1-ethyl-3-(3-dimethylaminopropyl)-carbodiimide hydrochloride (Nacalai Tesque) at 25 °C for 2 h, and carboxyl groups on the beads were activated. Fucoxanthinol was then immobilized onto the beads in the presence of DMAP (Nacalai Tesque) at 25 °C overnight. Unreacted residues on the beads were masked with 2-aminoethanol (Nacalai Tesque) at 25 °C for 4 h, and the resulting beads was stored at 4 °C.

**Purification and identification of fucoxanthinol-binding proteins.** HT-29 cells were lysed with NP-40 lysis buffer (50 mM Tris-HCl [pH 8.0], 150 mM NaCl, 1% NP-40, 1 mM DTT, and 0.5 mM PMSF) on ice for 30 min and centrifuged at $20,400 \times g$ for 10 min at 4 °C. The supernatants were used as the whole-cell extracts of HT-29 cells. The extracts were incubated with empty or fucoxanthinol-fixed beads at 4 °C for 4 h. The beads were then washed three times with binding buffer (50 mM Tris-HCl [pH 8.0], 150 mM NaCl, 0.1% NP-40). The proteins binding to the beads were eluted with Laemmli SDS sample buffer, subjected to 12% SDS-PAGE, and detected by silver staining. Each gel slice containing a fucoxanthinol-binding protein was subjected to in-gel digestion with Sequencing Grade Modified Trypsin (Promega, Madison, WI, USA). The resulting peptides were analyzed by Autoflex II (Bruker Daltonics, Billerica, MA, USA), and proteins were identified by peptide mass fingerprinting.

The recombinant His-tagged uS7 protein (His-uS7, ab137146, Abcam) was incubated with empty or fucoxanthinol-fixed beads at 4 °C for 4 h. The beads were then washed three times with binding buffer. His-uS7 binding to the beads was eluted with Laemmli SDS sample buffer, subjected to 12% SDS-PAGE, and detected by western blotting.

**RNAi.** HT-29 or SW480 cells were transfected with each siRNA using Lipofectamine RNAiMAX (Thermo Fisher Scientific). The following Stealth RNAi siRNAs (Thermo Fisher Scientific) were used: siuS7 #1 (HSS109357), 5'-GGAGCACC-GAUGAUGUGCAGAUCAA-3'; siuS7 #2 (HSS184431), 5'-GCCGCAA-CAACGGCAAGAAGCUCAU-3'; and a negative control siRNA (Cat. No. 12935-112). Only the RNA sequences of sense strands are shown.

**Cell fractionation.** HT-29 cells transfected with each siRNA were suspended in Buffer A (10 mM HEPES-NaOH [pH 7.9], 10 mM KCl, 0.1 mM EDTA, 1 mM DTT, 0.5 mM PMSF, and Phosphatase Inhibitor Cocktail) by gentle pipetting and incubated on ice for 15 min. A 10% solution of NP-40 was added to the suspension, and the suspension was vigorously vortexed for 10 s and centrifuged at $800 \times g$ for 1 min at 4 °C. The supernatants were collected as the cytoplasmic fraction of HT-29 cells. The pellets were suspended in Buffer C (20 mM HEPES-NaOH [pH 7.9], 400 mM NaCl, 1 mM EDTA, 1 mM DTT, 0.5 mM PMSF, and Phosphatase Inhibitor Cocktail). The suspension was vigorously rocked for 15 min at 4 °C and centrifuged at $20,400 \times g$ for 5 min at 4 °C. The supernatants were collected as the nuclear fraction of HT-29 cells.

**Cycloheximide chase assay.** HT-29 or SW480 cells were transfected with each siRNA. After 18 h, cells were treated with 20 μg/ml cycloheximide and sequentially collected. Cells were lysed with RIPA buffer (50 mM Tris-HCl [pH 8.0], 150 mM NaCl, 1% NP-40, 0.5% deoxycholic acid, 0.1% SDS, 1 mM DTT, 0.5 mM PMSF, and Phosphatase Inhibitor Cocktail) and subjected to a western blot analysis.

**Co-immunoprecipitation.** HT-29 cells were lysed with NETN buffer (20 mM Tris-HCl [pH 8.0], 100 mM NaCl, 0.5 mM EDTA, 0.5% NP-40, and 0.5 mM PMSF) at 4 °C for 30 min and centrifuged. The supernatants were incubated with normal rabbit IgG (sc-2027, Santa Cruz Biotechnology), anti-CDK2 (sc-163, Santa Cruz Biotechnology), CDK4 (sc-601, Santa Cruz Biotechnology), CDK6 (sc-177, Santa Cruz Biotechnology), cyclin D (Cat. No. 06-137, Merck KGaA, Darmstadt, Germany), or cyclin E (sc-198, Santa Cruz Biotechnology) antibodies at 4 °C overnight, and were then incubated with Protein G-conjugated FG beads® (TAS8848N1173, Tamagawa Seiki) at 4 °C for 2 h. Beads were washed three times with NETN buffer, and immunoprecipitates were eluted with Laemmli SDS sample buffer and subjected to a western blot analysis with Clean-Blot IP Detection Reagent-HRP (Thermo Fisher Scientific) instead of secondary antibodies.

**Preparation of GST-fused proteins.** A cDNA library from HT-29 cells was synthesized with Sepasol-RNA I Super G (Nacalai Tesque) and High-Capacity cDNA Reverse Transcription Kit (Applied Biosystems). The DNA fragment encoding human uS7 was then amplified by PCR using primers A (5'-AAAAGGATCCATGACCGAGTGGGAGACAG-3') and B (5'-AAAACTC-GAGTCAGCGGTTGGACTTGGCC-3') and subcloned into the expression vector pGEX-6P-1 (GE Healthcare). GST or the GST-fused uS7 (GST-uS7) protein was

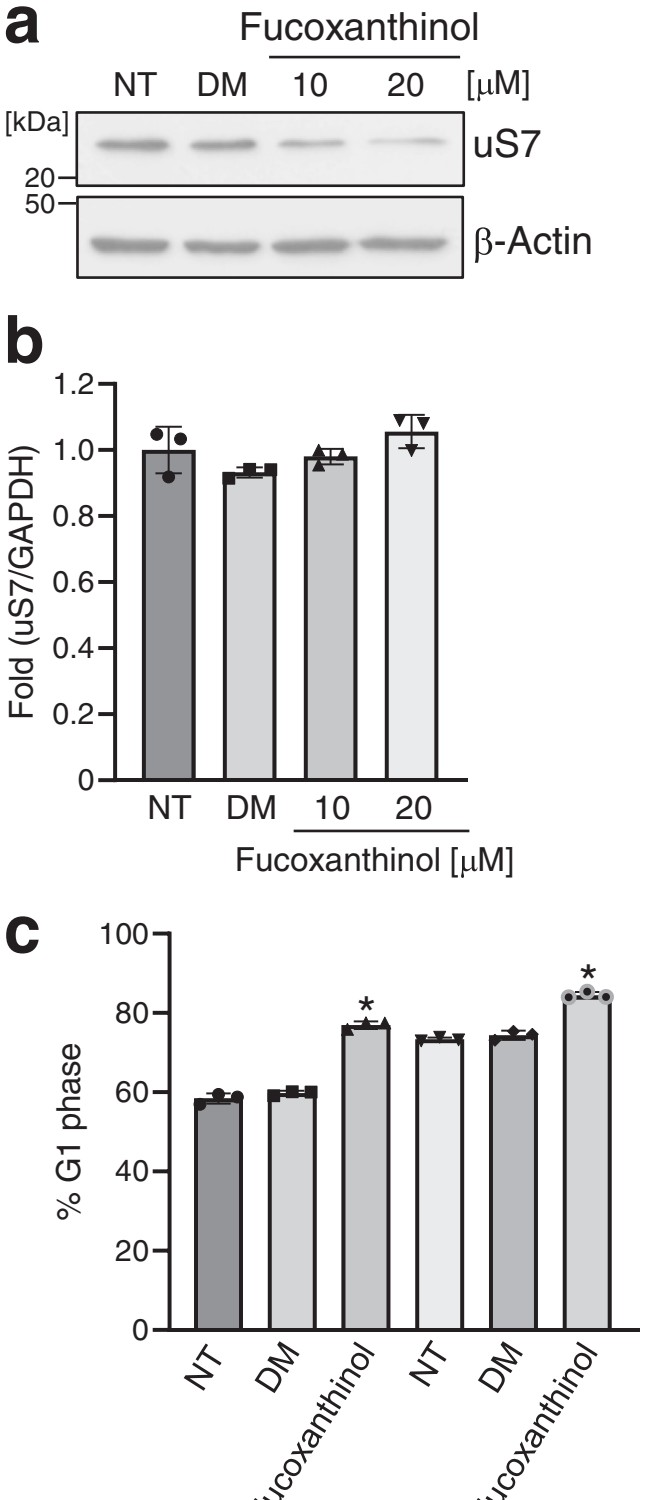

**Fig. 6 Fucoxanthinol decreases the uS7 protein, resulting in G1 cell cycle arrest.** HT-29 cells were treated with the indicated concentrations of fucoxanthinol for 6 h, and the protein (**a**) and mRNA (**b**) levels of uS7 were examined by western blotting and qRT-PCR analyses. **c** HT-29 cells were transfected with siCtrl or siuS7 #2. After 48 h, cells were incubated with 10 μM fucoxanthinol for 24 h, and the cell cycle was analyzed by flow cytometry. NT: non-treated, DM: 0.1% DMSO, Data are means ± S.D. ($n = 3$ biologically independent experiments). *$P < 0.05$ significantly different from NT.

expressed in BL21-CodonPlus (DE3)-RIPL *Escherichia coli* cells (Agilent Technologies, Santa Clara, CA, USA), and the lysates containing each protein were prepared by sonication.

**GST pull-down assay**. GST or the GST-uS7 protein was immobilized onto Glutathione Sepharose 4B beads (GE Healthcare) by incubating the *Escherichia coli* lysates expressing each protein with the beads at 4 °C for 2 h. The recombinant Myc-DDK-tagged CDK6 protein (OriGene Technologies) was then incubated with GST or GST-uS7-immobilized beads at 4 °C overnight. The beads were then washed three times with NETN buffer, and the proteins that bound to the beads were eluted with Laemmli SDS sample buffer and subjected to a western blot analysis.

**Docking simulation**. The structure of uS7 was derived from the human ribosome 80 S subunit structure (PDB code: 4UG0). The molecular docking simulation was performed using AutoDock Vina version 1.2.0[35]. To consider ligand flexibility, single bonds, except for the π-conjugation of fucoxanthinol, were allowed to rotate, and the structure of uS7 was kept rigid. A total of 100 poses of the complex of uS7 and fucoxanthinol were calculated and ranked according to the Vina scoring function[36]. The hydrophobicity of uS7 was calculated using the hydropathy index[37].

**MD simulations**. The microscopic state of the interaction between the uS7 protein and fucoxanthinol was examined using MD simulations. The uS7 protein was described using the Amber ff14SB force field. Fucoxanthinol was described using the General Amber Force Field[38] with restrained electrostatic potential (RESP) charges[39]. The partial charge of fucoxanthinol is shown in Supplementary Data 2. Water molecules were described using the TIP3P model[40]. The system contained the uS7 protein, a fucoxanthinol molecule, 26,922 water molecules, and 78 sodium and chloride ions, corresponding to 150 mM solution. To neutralize the net charge of the system, 14 chloride ions were added. These molecules were placed in a rhombic dodecahedron box with a lattice length of 10.68 nm. Temperature was maintained at 300 K using Langevin dynamics, and pressure was controlled at 1 atm using the Parrinello-Rahman barostat[41]. MD simulations were conducted using the GROMACS 2020.6 simulator[42]. A movie of the MD simulation trajectory of the uS7-fucoxanthinol complex was made by Visual molecular dynamics (VMD)[43].

**Statistics and reproducibility**. All experiments were performed at least two times with similar results. GraphPad Prism (version 9.3.0, GraphPad software, San Diego, CA, USA) and Microsoft Excel 2013 (Microsoft Corporation, Redmond, WA, USA) were used to plot and represent data. Data are shown as the mean ± standard deviation (S.D.) of the measurements of three independent experiments, and individual data points were plotted as symbols above the bars of the histograms. Statistical analyses were performed using a non-repeated measures ANOVA followed by Bonferroni correction using the Excel Statistical Program File ystat2013 (Igakutosho Shuppan Ltd., Saitama, Japan).

**Reporting summary**. Further information on research design is available in the Nature Research Reporting Summary linked to this article.

## Data availability

All data generated or analyzed during the present study are included in the article and its Supplementary Information file. Source data for the main figures can be found in Supplementary Data 1. All relevant data relating to the article are available from the corresponding author on reasonable request. Uncropped blots are provided as Supplementary Fig. 5-14 in the Supplementary Information file.

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

## Acknowledgements

We would like to thank Dr. Kazuhiro Yagita for his productive discussions and Dr. Tomoyuki Taniguchi for his technical support. This work was supported by JSPS KAKENHI Grant Numbers JP17H06398 and JP23689036.

## Author contributions

Y.I., Y.S., and T.S. conceived the project. Y.I., W.G., Y.A., and M.W. performed the experiments. Y.K. and T.K. performed docking and MD simulations. Y.I. and K.A. developed the method to immobilize a compound with a hydroxyl group onto FG beads®. M.K. provided important suggestions. Y.I. and T.S. wrote the manuscript.

## Competing interests

The authors declare no competing interests.
