## [Peer Review File · Communications Biology]

Reviewers' comments:

Reviewer #1 (Remarks to the Author):

In the manuscript "The natural product fucoxanthinol unveils ribosomal protein S5 mediated CDK6 stability" the Authors aimed to shed light on the mechanisms underlying the constant expression of cyclin-dependent kinases (CDKs). Specifically, they explored compounds that affect the expression of CDKs and found out the natural product fucoxanthinol as a regulator of CDK2, 4, and 6 at protein level. In addition, they identified ribosomal protein S5 (RPS5) as the primary binding protein of fucoxanthinol using magnetic affinity beads. Intriguingly, knockdown of RPS5 reduced CDK2, 4, and 6 levels in HT-29 cells, whereas only CDK6 was decreased in RPS5-depleted SW480 cells. RPS5 was involved in the protein stability of CDK6. Overall, these data indicate an extra-ribosomal function of RPS5 which stabilizes CDK6 protein.

Uncovering the mechanisms that regulate CDKs protein expression in cancer cells are of crucial importance for the development of anticancer therapeutic strategies based on CDK inhibitors. In this context, the topic of the manuscript is interesting and up-to-date.

The manuscript is written clearly and the presentation of results follows a coherent line. However, I have some comments on the current version of the manuscript to share with the Authors.

Introduction section:

- I recommend referring to ribosomal protein S5 as uS7 according to the new nomenclature for ribosomal proteins.

- The authors claimed that a number of ribosomal proteins are known to play a role outside the ribosome. In this context, few examples are given. Different recent studies have revealed that ribosomal proteins as ribosome-free forms are able to exert a variety of extra-ribosomal functions and some ribosomal proteins can impact the cell cycle depending on p53. There are recent related papers that might be mentioned/discussed too:

- Pecoraro A. et al. Ribosome Biogenesis and Cancer: Overview on Ribosomal Proteins. *Int J Mol Sci* (2021)

In particular, it has been demonstrated that some ribosomal proteins are able to modulate cell cycle progression at G1/S phase. There is a related paper that might be mentioned/discussed too: Pecoraro A. et al. Ribosomal protein uL3 targets E2F1 and Cyclin D1 in cancer cell response to nucleolar stress. *Sci Rep* (2019).

Results and Figures

- The Authors should provide uncropped full-length blots showed in Figure 1, 3, 4, 5.

- The Authors should add densitometric analyses of western blot signals showed in Figure 1C, 3D, 3E, 4C. It could be useful for a better interpretation of the results.

- Depletion of an individual ribosomal protein may result in a decrease in the level of the other ribosomal proteins belonging to the same ribosomal subunit, thus creating an unbalanced ribosome assembly pathway. What about the expression levels of other ribosomal proteins? Author could analyze at least the expression of another ribosomal protein.

- It is known that p21 regulates the progression of the cell cycle at the G1-S transition checkpoint. It would be of interest to verify the effect of Fucoxanthinol mediated down-regulation of the ribosomal protein S5 on p21 expression.

- At least, one key finding should be reproduced in condition of ribosomal protein S5 over-expression.

Discussion

- line 194-195: the Authors report "Depletion of RPS5 downregulated CDK2, 4, and 6 in HT-29 cells (Fig. 3d), whereas only CDK6 was reduced in RPS5-depleted". HT29 and SW480 cells are p53 mutated and p53 proficient cells, respectively. The Authors should comment these data in the context of recent findings indicating specific activities of a subset of ribosomal proteins depending on p53 status in the cell (Ribosomal Proteins Control or Bypass p53 during Nucleolar Stress. Russo A, Russo G. *Int J Mol Sci*. 2017, 18(1):140. doi: 10.3390/ijms18010140)

- Authors could add examples of other natural compound affecting ribosomal protein expression

and activity leading to cell cycle arrest as recently reported for S-adenosyl-L-methionine. There is a related paper that could be comment: Mosca L et al., Int J Mol Sci 2020 Dec 24;22(1):103. doi: 10.3390/ijms22010103.

Quality of English language:

The manuscript should be English proofread.

Reviewer #2 (Remarks to the Author):

The manuscript by Iizumi et al. shows that fucoxanthinol downregulates CDK2, CDK4 and CDK6 in the colon cancer cell line HT-29. Using fucoxanthinol immobilized to FG-beads®, the authors identified the ribosomal protein S5 (RPS5) as a major fucoxanthinol-binding protein. By the way, this manuscript also investigates the role of RPS5 in cell cycle control and show that RPS5 interact with CDK6 and mediated its stability. Inversely, knockdown of RPS5 induced G1 cell cycle arrest with downregulation of CDK6 in colon cancer cells.

This original study is of interest and scientifically sound. The manuscript is well written and the data support most of the conclusions. However, some points need to be addressed for further consideration.

Specific major concerns:

- In the results, the effect of fucoxanthinol on the cell cycle (Figure 1b) should be more detailed. Why the % of G1 (60%) decrease and the % of G2/M (~28%) increase with 20 µM fucoxanthinol compared to 10µM fucoxanthinol (79% and ~12%, respectively)?
- Do the authors have a hypothesis on how fucoxanthinol can bind to proteins like RPS5? Is there a specific binding site or structural motif? Could a binding model be performed (e.g. using PyMOL)?
- How the binding of fucoxanthinol to RPS5 can affect the cellular level of RPS5? Does fucoxanthinol binding activate degradation of RPS5? An additional experiment with MG132 should be performed to address whether the proteasome is involved in the fucoxanthinol-mediated degradation of RPS5.
- The effect of fucoxanthinol on the cellular level is the same for CDK2, CDK4 and CDK6 in HT-29 cells (Fig.1c). The downregulation of RPS5 in HT-29 cells lead to a similar pattern (Fig.3d), suggesting that fucoxanthinol binding to RPS5 is responsible of the observed downregulation of CDK. It is a bit different in another colon cancer cell line (SW480), where only CDK6 level is affected. This leads the authors to conclude that RPS5 only affect CDK6 level. But what is the effect of fucoxanthinol on the cell cycle of SW480 cells? The results of Figure 1 should be reproduced with the SW480 cell line.

Specific minor concerns:

- The title proposed for this manuscript is not very explicit and does not fully reflect its content. Please, revise it.
- FG-beads is a registered trademark and should therefore be noted as FG-beads® throughout the manuscript.
- Thank you for adding line numbers, it is so simple and makes the reviewers' work much easier. I don't understand why this is not an absolute requirement for submitting a manuscript.

Response to the Reviewers' suggestions and comments

We would like to thank the Reviewers for their thoughtful review and constructive comments on our manuscript. We performed additional experiments and revised the manuscript according to the comments provided and the editorial policies of Nature Research. We herein outline the changes made to the revised manuscript. In the revised manuscript, the changes according to the **Reviewers' comments** and **English proofreading** were highlighted in red and blue, respectively.

Reviewer #1

In the manuscript "The natural product fucoxanthinol unveils ribosomal protein S5 mediated CDK6 stability" the Authors aimed to shed light on the mechanisms underlying the constant expression of cyclin-dependent kinases (CDKs). Specifically, they explored compounds that affect the expression of CDKs and found out the natural product fucoxanthinol as a regulator of CDK2, 4, and 6 at protein level. In addition, they identified ribosomal protein S5 (RPS5) as the primary binding protein of fucoxanthinol using magnetic affinity beads. Intriguingly, knockdown of RPS5 reduced CDK2, 4, and 6 levels in HT-29 cells, whereas only CDK6 was decreased in RPS5-depleted SW480 cells. RPS5 was involved in the protein stability of CDK6. Overall, these data indicate an extra-ribosomal function of RPS5 which stabilizes CDK6 protein.

Uncovering the mechanisms that regulate CDKs protein expression in cancer cells are of crucial importance for the development of anticancer therapeutic strategies based on CDK inhibitors.

In this context, the topic of the manuscript is interesting and up-to-date.

The manuscript is written clearly and the presentation of results follows a coherent line. However, I have some comments on the current version of the manuscript to share with the Authors.

Introduction section:

(1) I recommend referring to ribosomal protein S5 as uS7 according to the new nomenclature for ribosomal proteins.

Thank you for informing us of the new nomenclature for ribosomal proteins. According to the new system for naming ribosomal proteins (Curr Opin Struct Biol. 2014 Feb;24:165-9. doi: 10.1016/j.sbi.2014.01.002), in which the new nomenclature for ribosomal proteins was described in detail, we changed the description "ribosomal

protein S5” to “ribosomal protein uS7” throughout the revised manuscript.

(2) *The authors claimed that a number of ribosomal proteins are known to play a role outside the ribosome. In this context, few examples are given. Different recent studies have revealed that ribosomal proteins as ribosome-free forms are able to exert a variety of extra-ribosomal functions and some ribosomal proteins can impact the cell cycle depending on p53. There are recent related papers that might be mentioned/discussed too:*

- Pecoraro A. et al. Ribosome Biogenesis and Cancer: Overview on Ribosomal Proteins. Int J Mol Sci (2021)

In particular, it has been demonstrated that some ribosomal proteins are able to module cell cycle progression at G1/S phase. There is a related paper that might be mentioned/discussed too: Pecoraro A. et al. Ribosomal protein uL3 targets E2F1 and Cyclin D1 in cancer cell response to nucleolar stress. Sci Rep (2019).

Thank you for informing us of the regulation of the cell cycle by several ribosomal proteins. To reinforce the description on the extraribosomal functions of ribosomal proteins, we described cell cycle regulation at the G1/S phase by uS15 and uL3 by citing the above-described papers (ref. 11 and 12 in the revised manuscript) in the Introduction section of the revised manuscript (lines 51-53).

Results and Figures

(3) *The Authors should provide uncropped full-length blots showed in Figure 1, 3, 4, 5.*

According to this comment, we added uncropped images of Western blots in all Figures as Supplementary Figures S5-14.

(4) *The Authors should add densitometric analyses of western blot signals showed in Figure 1C, 3D, 3E, 4C. It could be useful for a better interpretation of the results.*

According to this comment, the Western blot signals of revised Figures 1C, 4D, 4E, and 5C were quantified using the ImageJ software (Version 1.52a) and normalized by the value of β -actin, and we indicated the values of these signals below each blot. Since Western blots of the RB protein are evaluated by changes in molecular weight, we did not quantify these blots. Furthermore, the expression of p21 was not detected in non-treated cells (NT); therefore, Western blots of p21 were also not quantified.

(5) *Depletion of an individual ribosomal protein may result in a decrease in the level of the other ribosomal proteins belonging to the same ribosomal subunit, thus creating an*

unbalanced ribosome assembly pathway. What about the expression levels of other ribosomal proteins? Author could analyze at least the expression of another ribosomal protein.

According to this suggestion, we investigated the expression of ribosomal proteins uS4 (a component of the small ribosomal subunit) and uL3 (a component of the large ribosomal subunit) in HT-29 and SW480 cells in which ribosomal protein uS7 was knocked down. As shown in Supplementary Figure S3a, the knockdown of uS7 slightly decreased the expression of uS4 and uL3 in HT-29 cells. On the other hand, the knockdown of uS7 down-regulated the expression of uS4, but not uL3 in SW480 cells (Supplementary Fig. S3b). Therefore, the knockdown of uS7 may influence the expression of other components of the small ribosomal subunit in a cell line-specific manner. Further detailed analyses of interactional effects on other ribosomal proteins may clarify the overall mechanisms underlying the various bioactivities of fucoxanthinol. We described these results in the Results section of the revised manuscript (lines 132-138). Since the anti-ribosomal protein uS7 antibody (ab58345, Abcam) went out of production, the anti-uS7 antibody (ab210745, Abcam) was used in the revision. We described this in the Methods section of the revised manuscript (line 257). Since Dr. Motoki Watanabe also performed these experiments, we added him as an author (lines 4-5; line 487).

(6) *It is known that p21 regulates the progression of the cell cycle at the G1-S transition checkpoint. It would be of interest to verify the effect of Fucoxanthinol mediated down-regulation of the ribosomal protein S5 on p21 expression.*

According to the Reviewer's suggestion, we analyzed the expression of p21 in fucoxanthinol-treated or ribosomal protein S5 (uS7)-knocked down HT-29 cells. As shown in revised Figure 1c, fucoxanthinol induced p21 protein expression. On the other hand, the knockdown of uS7 did not induce p21 protein expression (revised Fig. 4d). These results suggest that the down-regulation of uS7 expression is not involved in the induction of p21 by fucoxanthinol. We described this in the Results section of the revised manuscript (lines 81-82; lines 125-126).

(7) *At least, one key finding should be reproduced in condition of ribosomal protein S5 over-expression.*

According to this comment, we initially evaluated the transfection efficacy to HT-29 cells using various transfection reagents and conditions with the GFP expression vector pEGFP-C1; however, transfection efficiency was very low (< 5%). We then

assessed transfection efficacy to SW480 cells, and the fluorescence of GFP was detected in approximately 60% of SW480 cells using Lipofectamine 3000 reagent (Thermo Fisher Scientific). Therefore, SW480 cells were transfected with the expression vector of the C-terminal Myc-DDK-tagged ribosomal protein S5 (uS7) (pCMV6-uS7-Myc-DDK, RC204595, OriGene Technologies) using this reagent for 48 h, and protein expression levels were analyzed by Western blotting. As shown in attached Figure A, the transient expression of the uS7-Myc-DDK protein was detected with anti-DDK and anti-uS7 antibodies. However, the exogenous expression of uS7 was much lower than the endogenous expression of uS7. Therefore, difficulties were associated with evaluating the effects of the overexpression of uS7 on CDK6 expression. We speculated the involvement of a mechanism that suppresses the exogenous expression of uS7 in SW480 cells, for example, the lack of a dedicated cytosolic ribosomal protein chaperone that protects newly synthesized uS7 from degradation, as reported in the above-described review (Pecoraro A. *et al.* Ribosome Biogenesis and Cancer: Overview on Ribosomal Proteins. *Int J Mol Sci.*, 2021).

Discussion

(8) *lane 194-195: the Authors report “Depletion of RPS5 downregulated CDK2, 4, and 6 in HT-29 cells (Fig. 3d), whereas only CDK6 was reduced in RPS5-depleted”. HT29 and SW480 cells are p53 mutated and p53 proficient cells, respectively. The Authors should comment these data in the context of recent findings indicating specific activities of a subset of ribosomal proteins depending on p53 status in the cell (Ribosomal Proteins Control or Bypass p53 during Nucleolar Stress.*

Russo A, Russo G. Int J Mol Sci. 2017, 18(1):140. doi: 10.3390/ijms18010140)

Thank you for informing us that some of the activities of ribosomal proteins depend on the p53 status. Although HT-29 and SW480 cells both carry the same dominant-negative mutation of p53 (R273H) (ref. 24 in the revised manuscript), p53 mutations in SW480 cells (R273H and P309S) are reported to retain some normal p53 functions (ref. 25 in the revised manuscript). This different status of p53 between HT-29 and SW480 cells may be one of the mechanisms underlying the different regulation of CDK2 and 4 between them. We described this by citing the above-described review (ref. 23 in the revised manuscript) in the Discussion section of the revised manuscript (lines 186-190).

(9) *Authors could add examples of other natural compound affecting ribosomal protein expression and activity leading to cell cycle arrest as recently reported for*

S-adenosyl-L-methionine. There is a related paper that could be comment: Mosca L et al., *Int J Mol Sci* 2020 Dec 24;22(1):103. doi: 10.3390/ijms22010103.

In the study introduced by the Reviewer, ribosomal protein uL3 was reported to prevent S cell cycle arrest induced by S-adenosyl-L-methionine. We also previously demonstrated that the natural product apigenin induces G2/M cell cycle arrest by directly binding to ribosomal protein S9 (uS4) (ref. 28 in the revised manuscript). Some relationships may exist between ribosomal proteins and the bioactivities of natural products. We described this by citing the suggested study (ref. 29 in the revised manuscript) in the Discussion section of the revised manuscript (lines 208-214).

(10) Quality of English language:

The manuscript should be English proofread.

The revised manuscript was subjected to English proofreading.

Reviewer #2

The manuscript by Iizumi et al. shows that fucoxanthinol downregulates CDK2, CDK4 and CDK6 in the colon cancer cell line HT-29. Using fucoxanthinol immobilized to FG-beads®, the authors identified the ribosomal protein S5 (RPS5) as a major fucoxanthinol-binding protein. By the way, this manuscript also investigates the role of RPS5 in cell cycle control and show that RPS5 interact with CDK6 and mediated its stability. Inversely, knockdown of RPS5 induced G1 cell cycle arrest with downregulation of CDK6 in colon cancer cells.

This original study is of interest and scientifically sound. The manuscript is well written and the data support most of the conclusions. However, some points need to be addressed for further consideration.

According to Reviewer #1's suggestion that "I recommend referring to ribosomal protein S5 as uS7 according to the new nomenclature for ribosomal proteins", we referred to ribosomal proteins according to the new system for naming ribosomal proteins (*Curr Opin Struct Biol.* 2014 Feb;24:165-9. doi: 10.1016/j.sbi.2014.01.002).

Specific major concerns:

(1) *In the results, the effect of fucoxanthinol on the cell cycle (Figure 1b) should be more detailed. Why the % of G1 (60%) decrease and the % of G2/M (~28%) increase with 20 µM fucoxanthinol compared to 10 µM fucoxanthinol (79% and ~12%.*

respectively)?

As pointed out by the Reviewer, a high concentration of fucoxanthinol (20 μM) induced G2/M and G1 arrest in HT-29 cells (Fig. 1b). As shown in Supplementary Figure 1b, a high concentration of fucoxanthinol (10 μM) induced G2/M arrest, but not G1 arrest in SW480 cells. Since high concentrations of fucoxanthin, the precursor of fucoxanthinol, have been shown to down-regulate the expression of cyclin B1, a positive regulator of G2/M transition, and induce G2/M arrest (ref. 20 in the revised manuscript), high concentrations of fucoxanthinol may also induce G2/M arrest by down-regulating cyclin B1 expression. We described this in the Results section of the revised manuscript (lines 79-80; lines 82-88).

(2) *Do the authors have a hypothesis on how fucoxanthinol can bind to proteins like RPS5? Is there a specific binding site or structural motif? Could a binding model be performed (e.g. using PyMOL)?*

According to this suggestion, the microscopic state of the interaction between RPS5 (uS7) and fucoxanthinol was examined using docking and molecular dynamics (MD) simulations. We generated 100 poses of the complex of uS7 and fucoxanthinol by docking simulations (revised Fig. 3a). In the top 2 poses, fucoxanthinol bound to almost the same position on uS7, and the allenic group of fucoxanthinol was located at a hydrophobic pocket of uS7 (revised Fig. 3b). To investigate the stability of the complex structure, we performed MD simulations. One hundred ns simulations were performed three times. In all simulations, fucoxanthinol remained at the same position of uS7 (Supplementary Video 1), suggesting that it strongly binds to the hydrophobic pocket of uS7. We described these results in the Results, Methods, and Figure legends sections of the revised manuscript (lines 107-115; lines 336-358; lines 519-524). Since Drs. Yoichi Kurumida and Tomoshi Kameda performed these simulations and analyses, we added them as authors (line 5; line 488).

(3) *How the binding of fucoxanthinol to RPS5 can affect the cellular level of RPS5? Does fucoxanthinol binding activate degradation of RPS5? An additional experiment with MG132 should be performed to address whether the proteasome is involved in the fucoxanthinol-mediated degradation of RPS5.*

According to the Reviewer's comment, we examined the involvement of the proteasome in the reduction of RPS5 (uS7) by fucoxanthinol using MG132 and epoxomicin (4381-v, Peptide Institute), which is a potent and selective proteasome inhibitor (Proc Natl Acad Sci USA. 1999 Aug 31;96(18):10403-8. doi:

10.1073/pnas.96.18.10403). As shown in attached Figure B, neither MG132 nor epoxomicin inhibited the reduction of uS7 by fucoxanthinol. Unexpectedly, in the absence of fucoxanthinol, MG132 and epoxomicin both down-regulated uS7 expression, indicating that the inhibition of the proteasome decreases uS7 expression. Unfortunately, we were unable to investigate whether fucoxanthinol decreases uS7 in a manner that is dependent on the proteasome.

(4) *The effect of fucoxanthinol on the cellular level is the same for CDK2, CDK4 and CDK6 in HT-29 cells (Fig. 1c). The downregulation of RPS5 in HT-29 cells lead to a similar pattern (Fig. 3d), suggesting that fucoxanthinol binding to RPS5 is responsible of the observed downregulation of CDK. It is a bit different in another colon cancer cell line (SW480), where only CDK6 level is affected. This leads the authors to conclude that RPS5 only affect CDK6 level. But what is the effect of fucoxanthinol on the cell cycle of SW480 cells? The results of Figure 1 should be reproduced with the SW480 cell line.*

According to this comment, we examined the effects of fucoxanthinol in SW480 cells. As shown in Supplementary Figure S1a and b, fucoxanthinol inhibited the proliferation of SW480 cells and induced G1 arrest at lower concentrations than against HT-29 cells. In addition, high concentrations of fucoxanthinol induced G2/M arrest. As shown in Supplementary Figure S1c, fucoxanthinol down-regulated CDK2, 4, and 6 expression, similar to HT-29 cells. Since the knockdown of RPS5 (uS7) did not down-regulate CDK2 or 4 expression in SW480 cells (revised Fig. 4e), fucoxanthinol may have the ability to decrease the expression of CDK2 and 4 independently of binding to uS7. We described this in the Results section of the revised manuscript (lines 82-89).

Specific minor concerns:

(5) *The title proposed for this manuscript is not very explicit and does not fully reflect its content. Please, revise it.*

According to this suggestion, we changed the title of this manuscript to “Stabilization of CDK6 by ribosomal protein uS7, a target protein of the natural product fucoxanthinol”.

(6) *FG-beads is a registered trademark and should therefore be noted as FG-beads® throughout the manuscript.*

As indicated by the Reviewer, we noted FG beads as FG beads®.

(7) *Thank you for adding line numbers, it is so simple and makes the reviewers' work much easier. I don't understand why this is not an absolute requirement for submitting a manuscript.*

Thank you for your positive comments. However, we did not add line numbers. The online submission system of *Communications Biology* automatically added line numbers.

Reviewers' comments:

Reviewer #1 (Remarks to the Author):

The authors have answered my questions improving the manuscript.
I have no further comments to the content of the revised version of the manuscript.

Reviewer #2 (Remarks to the Author):

In the revised version of their manuscript, the authors have added several additional experiments according to reviewers' request. The molecular docking data showing the direct interaction of fucoxanthinol and uS7 protein are original and greatly improve our comprehension of fucoxanthinol molecular mechanisms. Although not conclusive, I acknowledge the authors for performing proteasome inhibitors experiments.

However, some minor points remain to be clarified:

1) In Figures 1 and S1, what does the presence of symbols (circle, squares, triangles) above the bars of the histograms mean? This seems to be redundant with the color coding used. Please redraw these histograms.

2) The first paragraph of the results section needs to be rewritten in a less confusing manner. The authors should identify the main points of the results presented for the 2 cell lines used. Thus, they should first describe the effect of fucoxanthinol on cell viability for both HT-29 (Fig. 1a) and SW480 (Fig. S1a) and pinpoint the higher sensibility of the W480 cell line. Then, the results of cell cycle analysis should be presented for both cell lines, highlighting the similarity of the effect between the two cell lines despite the shift in effective concentration. It is important to note that increasing the concentration of fucoxanthinol lead a concentration-dependent increase in G1-arrested cells and further increase in fucoxanthinol finally lead to shift from a G1 arrest to a G2/M arrest in both cell lines.

3) However, how cells, which are blocked in G1 phase in the presence of low concentration of fucoxanthinol (i.e. <10 or <5 μM in HT-29 or SW480 cells, respectively) can pass this same G1 phase and be blocked only in G2/M when a high concentration (i.e. 20 or 10 μM in HT-29 or SW480 cells, respectively) is used remains difficult to understand. Moreover, this is not explained by an additional decrease in the number of cells in S phase. Indeed, if the cells start to be arrested in G1, we cannot expect to have an increase of cells in G2/M! Do the authors have any explanation for this surprising effect? If yes please add it to the discussion.

Response to the Reviewers' suggestions and comments

Reviewer #1 (Remarks to the Author):

The authors have answered my questions improving the manuscript.

I have no further comments to the content of the revised version of the manuscript.

Thank you very much for your constructive suggestions. Owing to the suggestions, our manuscript has been much improved.

Reviewer #2 (Remarks to the Author):

In the revised version of their manuscript, the authors have added several additional experiments according to reviewers' request. The molecular docking data showing the direct interaction of fucoxanthinol and uS7 protein are original and greatly improve our comprehension of fucoxanthinol molecular mechanisms. Although not conclusive, I acknowledge the authors for performing proteasome inhibitors experiments.

Thank you very much for your constructive suggestions. In particular, by performing docking and molecular dynamics simulations as you suggested, we are happy to add new strong data and enhance our manuscript.

However, some minor points remain to be clarified:

1) *In Figures 1 and S1, what does the presence of symbols (circle, squares, triangles) above the bars of the histograms mean? This seems to be redundant with the color coding used. Please redraw these histograms.*

According to the editorial policies of Nature Research, individual data points were plotted as symbols above the bars of the histograms using GraphPad Prism (version 9.3.0). We described this in the Methods section of the revised manuscript (lines 371-372). Since the color coding of the bars was redundant with the symbols, we redrew these histograms without color coding.

2) *The first paragraph of the results section needs to be rewritten in a less confusing manner. The authors should identify the main points of the results presented for the 2 cell lines used. Thus, they should first describe the effect of fucoxanthinol on cell viability for both HT-29 (Fig. 1a) and SW480 (Fig. S1a) and pinpoint the higher sensibility of the SW480 cell line. Then, the results of cell cycle analysis should be*

presented for both cell lines, highlighting the similarity of the effect between the two cell lines despite the shift in effective concentration. It is important to note that increasing the concentration of fucoxanthinol lead a concentration-dependent increase in G1-arrested cells and further increase in fucoxanthinol finally lead to shift from a G1 arrest to a G2/M arrest in both cell lines.

Thank you for your careful suggestion to improve the first paragraph of the Results section. According to this suggestion, we rewrote this paragraph in a less confusing manner by describing the results of HT-29 and SW480 cells in parallel (lines 78-88). The discussion about G2/M arrest induced by fucoxanthinol was moved to the Discussion section of the revised manuscript (lines 206-213).

3) *However, how cells, which are blocked in G1 phase in the presence of low concentration of fucoxanthinol (i.e. <10 or <5 μM in HT-29 or SW480 cells, respectively) can pass this same G1 phase and be blocked only in G2/M when a high concentration (i.e. 20 or 10 μM in HT-29 or SW480 cells, respectively) is used remains difficult to understand. Moreover, this is not explained by an additional decrease in the number of cells in S phase. Indeed, if the cells start to be arrested in G1, we cannot expect to have an increase of cells in G2/M! Do the authors have any explanation for this surprising effect? If yes please add it to the discussion.*

Because these experiments are performed under normal culture conditions, each cell is in the G1, S, or G2/M phase when fucoxanthinol is added. For example, as shown in Figure 1b, non-treated (NT) HT-29 cells, which were not arrested, were in the G1 (about 40%), S (about 40%), or G2/M (about 20%) phase. Because low concentrations of fucoxanthinol induced only G1 arrest, most of the cells were arrested in the G1 phase. On the other hand, a high concentration of fucoxanthinol (20 or 10 μM in HT-29 or SW480 cells, respectively) also induced G2/M arrest, presumably by down-regulating cyclin B1 similarly to fucoxanthin, the precursor of fucoxanthinol (ref. 27 in the revised manuscript). When G2 arrest is induced, the cells which are in at least the S and G2 phases must be arrested in the G2/M phase, but not in the G1 phase. Because a high concentration of fucoxanthinol induced both G1 and G2/M arrest, the populations in both the G1 and G2/M phases might be large. We described this in the Discussion section of the revised manuscript (lines 206-213).